# Effects of Copper Content on the Microstructural, Mechanical and Tribological Properties of TiAlSiN–Cu Superhard Nanocomposite Coatings

**Sung-Bo Heo** [1,2], **Wang Ryeol Kim** [1], **Jun-Ho Kim** [1], **Su-Hyeon Choe** [1,2], **Daeil Kim** [2,*], **Jae-Hun Lim** [3] **and In-Wook Park** [1,*]

1   Dongnam Division, Korea Institute of Industrial Technology (KITECH), Yangsan 50623, Republic of Korea
2   School of Materials Science and Engineering, University of Ulsan, Ulsan 44610, Republic of Korea
3   BMT Co., Ltd., Yangsan 50568, Republic of Korea
*   Correspondence: dkim84@ulsan.ac.kr (D.K.); ipark@kitech.re.kr (I.-W.P.)

**Abstract:** The effects of the Cu content on the microstructural, mechanical and tribological properties of the TiAlSiN–Cu coatings were investigated in an effort to improve the wear resistance with a good fracture toughness for cutting tool applications. A functionally graded TiAlSiN–Cu coating with various copper (Cu) contents was fabricated by a filtered cathodic arc ion plating technique using four different (Ti, TiAl$_2$, Ti$_4$Si, and Ti$_4$Cu) targets in an argon-nitrogen atmosphere. The results showed that the TiAlSiN–Cu coatings are a nanocomposite consisting of (Ti,Al)N nano-crystallites (~5 to 7 nm) embedded in an amorphous matrix, which is a mixture of TiO$_x$, AlO$_x$, SiO$_x$, SiN$_x$, and CuO$_x$ phase. The addition of Cu atoms into the TiAlSiN coatings led to the formation of an amorphous copper oxide (CuO$_x$) phase in the coatings. The maximum nanohardness (H) of ~46 GPa, H/E ratio of ~0.102, and adhesion bonding strength between coating and substrate of ~60 N (L$_{C2}$) were obtained at a Cu content ranging from 1.02 to 2.92 at.% in the TiAlSiN–Cu coatings. The coating with the lowest friction coefficient and best wear resistance was also obtained at a Cu content of 2.92 at.%. The formation of the amorphous CuO$_x$ phase during coating growth or sliding test played a key role as a smooth solid-lubricant layer, and reduced the average friction coefficient (~0.46) and wear rate (~10 × 10$^{-6}$ mm$^3$/N·m).

**Keywords:** TiAlSiN–Cu; copper addition; CuO$_x$; functionally graded coatings; superhard nanocomposite; wear resistance

## 1. Introduction

In the last few decades, various ternary coatings based on Ti–X–N and Cr–X–N systems have been widely explored with various deposition techniques and used in cutting tools as wear resistance protection due to their high hardness, high chemical stability, and excellent oxidation resistance [1,2]. However, the ternary Ti–X–N and Cr–X–N coatings prepared by a conventional process with low plasma ion energies or ion bombardments often showed a columnar structure and have a high friction coefficient [3]. In the various hard coating processes, the cathodic arc ion plating technique has attractive properties for preparing hard coatings, such as good adhesion and high deposition rates, and is characterized by a high ionization coupling with high plasma ion energies and a high current density compared with other conventional processes [4].

Recently, nanocomposite coatings are normally formed from ternary or higher order systems and comprise at least two immiscible phases: two nanocrystalline phases or, more commonly, an amorphous phase surrounding the nanocrystallites of a secondary phase. The most interesting and extensively investigated nanocomposite coatings are ternary, quaternary or even more complex systems, with nanocrystalline (nc-) grains of hard transition metal-nitrides, carbides, borides, oxides, or silicides surrounded by amorphous (a-)

matrices [2]. In addition, there have been extensive efforts to extend the ternary coating systems to quaternary ones in order to combine the favorable properties of each ternary system. In our previous studies, various quaternary coating systems, e.g., TiAlSiN [5], TiBCN [6], CrAlSiN [7], CrSiCN [8], CrAlMoN [9], and AlCrVN [10], etc., have been systematically investigated in terms of a multi-functional nanocomposite coating consisting of nano-sized metal nitride (referred to as *nc*-MeN) hard phases (e.g., TiN, TiC, $TiB_2$, CrN, etc) and an amorphous(*a*-) matrix (e.g., $a$-$Si_3N_4$, $a$-BN, $a$-C, etc). The incorporation of alloying elements affects the grain size, morphology, texture, chemical and phase compositions of the nanocomposite coatings [11,12]. Alloying elements also change the volume fractions of the nanocrystalline and amorphous; all these factors strongly influence the obtained mechanical and tribological properties of the coatings [7,12]. It was confirmed that most of the above quaternary coatings showed a superior hardness (>40 GPa, i.e., superhardness), excellent oxidation resistance (>900 °C), and good tribological properties [5,7]. Despite the combination of excellent properties, it was also found that they have a limitation or disadvantage regarding coating toughness [13,14]. High internal residual stress in the coatings often makes coated tools to lower tool lifetime during which they are applied to cutting machining [15].

More recently, in order to enhance the coating toughness, some soft metals, e.g., V, Ag, Cu, Mo, etc., have been added as a free or second phase into the grains or grain-boundaries and formed an adaptive nanocomposite structure such as nc-MeN/metal or amorphous phase [16–18]. Thus, these approaches and developments not only enhanced the fracture toughness of the coatings by adding a small amount of soft metals, but also provided a lubricating effect by the out-diffusion of soft metals at elevated temperatures [17,18]. These lubrication properties with the soft metals are due to their low shear strength and high plasticity. Soft metals such as Cu have been added in Mo–N coatings to form nanocomposite microstructures to achieve better tribological performance [19–21]. The incorporation of Cu into TiAlN coatings has also been studied [22]. According to the report, TiAlN–Cu coatings with about 1.3 at.% Cu can exhibit excellent mechanical properties and can demonstrate an excellent cutting performance. Yi et al. also found [23] that the addition of Cu(~1.3 at.%) into the AlTiN coatings resulted in a decrease in its grain size and hardness, and their turning experiments showed that the additive Cu effectively decreased the turning force, thereby extending tool lifetimes at various cutting speeds. However, few studies have been reported on the effect of Cu addition into the quinary coating systems [24,25], especially for the deposition of nanocomposites with superhard coatings.

Furthermore, in order to counteract brittleness and improve fracture toughness with high adhesion of the coatings, a functionally graded coating's (FGC's) architecture has been explored. For example, a substrate/Ti–TiN–TiCN–TiC–DLC (outside layer) graded coating for cutting tools was reported by Voevodin et al. [26]. The FGC's approach can be combined with multi-layered nanocomposite architecture to further enhance the coating fracture toughness and coating performance during cutting applications [27,28].

Therefore, in this work, quinary TiAlSiN–Cu coatings with the FGC's architecture were fabricated by the cathodic arc ion plating process with various Cu contents. The effects of the Cu content on the microstructural, mechanical and tribological properties of the TiAlSiN–Cu coatings were investigated in an effort to improve the wear resistance with good fracture toughness for cutting tool applications.

## 2. Experimental Procedure

### 2.1. Coating Deposition

TiAlSiN–Cu coatings were deposited onto silicon wafers for microstructural evaluations and WC–Co substrates for mechanical and wear tests by a filtered cathodic arc ion plating system with argon-nitrogen reactive gases. As shown in Figure 1, four different targets ($TiAl_2$, $Ti_4Si$, and $Ti_4Cu$ alloying targets and pure Ti target) were designed to prepare the TiAlSiN–Cu coatings with various compositions, similar to our previous study for the design of CrAlTiN–Si nanocomposite coatings [29]. A typical deposition condition

is summarized in Figure 1b with a cross-sectional SEM image of the TiAlSiN–Cu coating (Figure 1a). Prior to deposition, all substrates were ultrasonically cleaned in acetone, ethanol, and de-ionized water for 10, 15 and 10 min, respectively. Substrates were further cleaned in the deposition chamber by ion bombardment using a bias voltage of $-350$ V under an argon atmosphere of 8.8 Pa (65 mTorr) for 30 min. Next, functionally graded coatings (FGC's) and substrate/Ti/TiN/TiAlN/TiAlSiN/TiAlSiN–Cu multi-layers were fabricated, as shown in Figure 1a. Ti/TiN layers were deposited, as an adhesion layer of about 0.3 μm in thickness, to have a good bond strength between the adhesion layer and the WC–Co substrate. The supporting TiAlN/TiAlSiN layers for good fracture toughness of the coatings were then deposited at about 1.25 μm in thickness. The main TiAlSiN–Cu layer, finally, was deposited with a thickness of approximately 2.0 μm as a functional coating layer with characteristics of high hardness and excellent wear resistance. Detailed deposition conditions for each layer were listed in the table, as shown in Figure 1b.

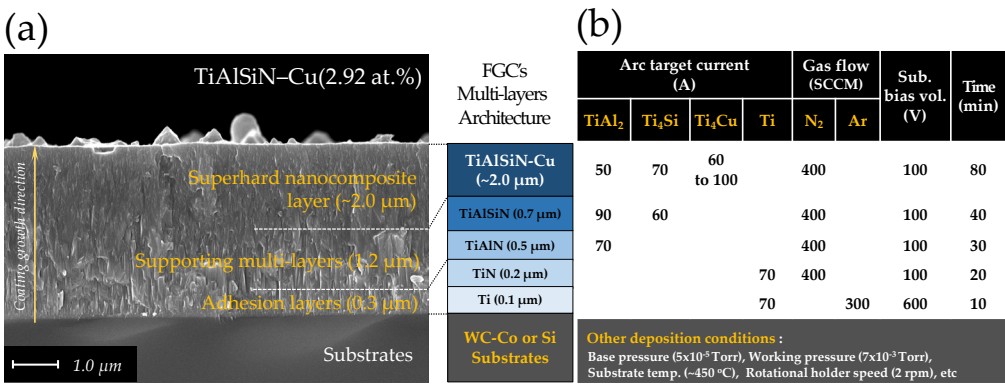

**Figure 1.** (**a**) Cross-sectional SEM micrograph for the functionally graded TiAlSiN–Cu coatings with multi-layers architecture and (**b**) typical depositions condition with various target powers by cathodic arc ion plating system with argon-nitrogen reactive gases.

## 2.2. Coating Characterisations

The thickness and surface morphology of the coatings were measured using a stylus profilometer (α-STEP, Tencor-10, KLA, Milpitas, CA, USA) and field emission scanning electron microscope (FE-SEM, JSM–7000F, JEOL, Tokyo, Japan), respectively. The chemical compositions were characterized by an electron probe micro-analyzer (EPMA, JXL-8100, JEOL, Tokyo, Japan). X-ray photoelectron spectroscopy (XPS, ESCALAB 250, Thermo Fisher Scientific, Waltham, MA, USA) analysis was performed with a monochromatic Al-K$_\alpha$ X-ray source to characterize the chemical bonding status of the TiAlSiN–Cu coatings. The atomic force microscope (AFM, Nanoscope IIIa module, Digital Instruments, Santa Barbara, CA, USA) consisting of a sharp tip (20 nm diameter) attached to a stiff cantilever was used to measure the surface roughness of the coatings. A high resolution transmission electron microscope (HRTEM, JEM-2200FS, JEOL, Tokyo, Japan) operated at 200 kV was used to examine the cross-sectional microstructure of the coatings. Cross-section TEM samples were prepared from the TiAlSiN–Cu coatings deposited on WC-Co samples using a focused ion beam (FIB, MI4050, Hitachi, Tokyo, Japan) with a Ga$^+$ liquid metal needle ion source and acceleration voltage of 30 kV setting in 5 kV steps. Relevant dark-field TEM (DF-TEM) images, selected-area electron diffraction (SAED) patterns, and high-resolution TEM (HRTEM) images were obtained to analyze the structural information. An inverse fast Fourier transform (IFFT) image was also obtained using a Gartan software program (Digital Micrograph TM, Gartan Inc., Warrendale, PA, USA).

The hardness and Young's modulus of the coatings were measured using a nanoindenter (Nanoindentation, NHT2, Anton Paar, Graz, Austria) equipped with a Berkovich diamond indenter (elastic modulus E = 1140 GPa, Poisson ratio ν = 0.07 and tip radius = 100 nm) was used to obtain the values of nanohardness (H) and Young's modulus (E) of the TiAlSiN–Cu coatings. The Berkovich tip was calibrated with a fused quartz reference sample. The

maximum indentation depths were controlled to be less than 10% of the film thickness to avoid the substrate effect. The effective Young's modulus ($E^* = E/(1 - \nu^2)$, where $\nu$ is a Poisson ratio of 0.25 for TiN-based coatings) of the coatings were calculated from the obtained Young's modulus (E) data. The adhesion strength between the coating and substrate was measured using an adhesion tester (Scratch tester, JLST022, J&L Tech Co., Ansan, Korea). The adhesive failure was classified into four modes in the scratch tests under progressive load (0 to 120 N). The first critical load ($L_{C1}$) was referred to as the semicircular cracks inside the scratch track, whereas the second critical load ($L_{C2}$) was referred to as the adhesive chipping at track edges. The critical loads $L_{C3}$ and $L_{C4}$ were referred to as the initial failure of coatings and total failure of coatings with exposed substrates, respectively. A conventional ball-on-disc friction and wear instrument was used to characterize the dry friction and wear performance of the TiAlSiN–Cu coatings. The sliding wear and tribological tests were conducted against Inconel balls (3.0 mm in diameter) with a sliding speed of 100 rpm, a load of 3 N in ambient room air (~25 °C, relative humidity: 40~50%), and for sliding distances up to 200 m. The surface morphology of the wear track was investigated by a field emission scanning electron microscope (FE-SEM, JSM–7000F, JEOL, Tokyo, Japan). To measure the wear volume, the three-dimensional profiles of the wear tracks were obtained using a white light interferometer (3D profiler, Contour GT-X3, Bruker, Billerica, MA, USA). The wear rates of the coatings were calculated as the wear volume per normal load and sliding distance ($mm^3$/N·m).

## 3. Results and Discussion

### 3.1. Chemical Compositions and Microstructure

The compositions of TiAlSiN–Cu coatings by EPMA analysis according to the change of target power were summarized in Table 1. With an increase in target current from 60 A to 100 A of $Ti_4Cu$ during deposition, Cu content in the TiAlSiN–Cu coatings was gradually incorporated from 1.01 to 4.74 at.%. The Ti content also linearly increased from 24.67 to 34.16 at.%. However, the Al and Si contents exhibited an opposite tendency, which decreased sharply from 15.24 to 6.01 at.% and from 5.04 to 2.12 at.%, respectively. The N and O contents remained almost constant, which varied in the small range of around 49 at.% and 4.8 at.%, respectively. The oxygen source seems to be delivered from the target or chamber as an impurity [6].

**Table 1.** Compositions of the TiAlSiN–Cu coatings fabricated by different target powers.

| Sample ID | Target Current (A) | | | Coating Composition (at.%) by EPMA | | | | | | Thickness (µm) |
|---|---|---|---|---|---|---|---|---|---|---|
| | $TiAl_2$ | $Ti_4Si$ | $Ti_4Cu$ | Ti | Al | Si | N | Cu | [1] O | |
| TiAlSiN–Cu(4.74 at.%) | 50 | 70 | 100 | 34.16 | 6.01 | 2.12 | 48.08 | 4.74 | 4.89 | 3.45 |
| TiAlSiN–Cu(3.85 at.%) | 50 | 70 | 90 | 31.12 | 8.09 | 2.91 | 49.01 | 3.85 | 5.02 | 3.51 |
| TiAlSiN–Cu(2.92 at.%) | 50 | 70 | 80 | 29.42 | 10.01 | 3.05 | 49.74 | 2.92 | 4.86 | 3.56 |
| TiAlSiN–Cu(1.89 at.%) | 50 | 70 | 70 | 26.62 | 13.56 | 4.09 | 48.91 | 1.89 | 4.94 | 3.62 |
| TiAlSiN–Cu(1.01 at.%) | 50 | 70 | 60 | 24.67 | 15.24 | 5.04 | 49.23 | 1.01 | 4.81 | 3.58 |

[1] Oxygen source was delivered from the target or chamber as an impurity.

Figure 2 shows the cross-sectional TEM images, corresponding SAED pattern, and IFFT image of the TiAlSiN–Cu(2.92 at.%) coating. In Figure 2a,b, the coating presented a dense structure along the growth direction in the dark-field TEM image, and the corresponding SAED pattern showed a poly-crystalline structure with (111), (200), (220), (311) and (222) crystal planes. All of the crystal planes were confirmed to have a crystalline (Ti,Al)N solid-solution phase. No diffraction ring or spot for Cu phase can be observed. Therefore, it can be suggested that the incorporated Cu atoms existed as an amorphous phase in the TiAlSiN–Cu coatings. A similar result has been found in the Mo–Cu–V–N coating with relatively high Cu content (~9 at.%) [30]. On the other hand, Shi et al. found the existence of a crystalline Cu phase with Cu (111) and Cu (200) planes in the Ti–Al–Si–Cu–N film [24].

In Figure 2c,d, a high-resolution TEM image and its inverse fast Fourier transform (IFFT) image exhibited a nanocomposite structure consisting of a large amount of nano-crystallites (*nc*-) and a thin layer of amorphous (*a*-) phases. The lattice spacing of the ordered fringes is 0.242 nm, which is smaller than that ($d_{TiN(111)} = 0.244$) of the standard TiN. Ti–Al–N that have been known as a solid-solution, in which Al atoms were substituted for Ti lattice sites up to 60 at.% [31]. The IFFT image further revealed that the TiAlSiN–Cu(2.92 at.%) coating has a nanocomposite structure, in which the crystallites exhibited regular and spherical shapes with a size ranging from 5 to 7 nm in an amorphous matrix.

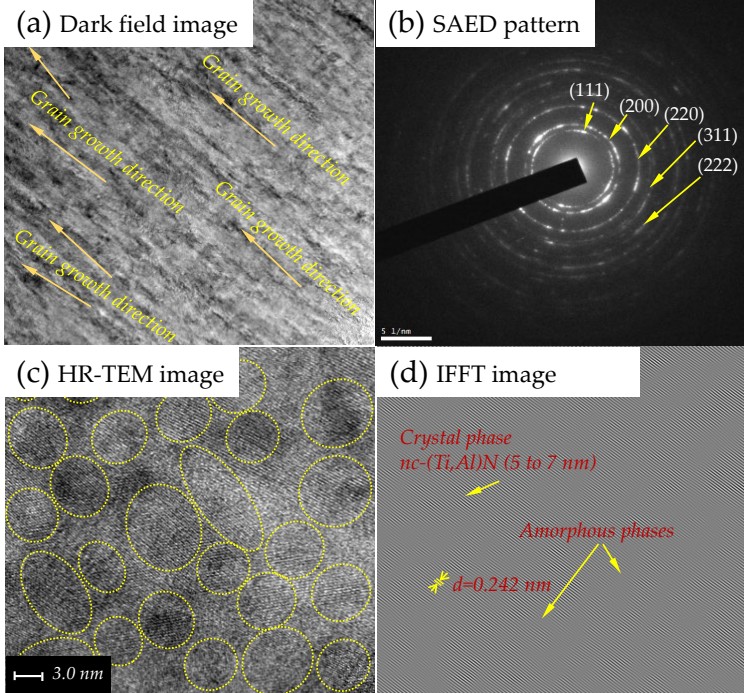

**Figure 2.** Cross-sectional TEM images of the TiAlSiN–Cu(2.92 at.%) coating: (**a**) dark field TEM image, (**b**) SAED pattern, and (**c**) HR-TEM image (yellow circles are nano-crystallites) and (**d**) corresponding inverse fast Fourier transform (IFFT) image.

To confirm the amorphous phases with detailed bonding status for each element in the coatings, an XPS analysis was applied to the TiAlSiN–Cu(2.92 at.%) coating, and the results are shown in Figure 3. The sample was referenced to the C 1s peak at a binding energy of 284.5 eV [32]. As shown in Figure 3a for Ti 2p, four peaks were observed. The main peaks at 457.4 and 464.5 eV with a large percentage of area corresponded to the TiN compound. There was also a small contribution from the titanium oxide ($TiO_x$) phase at 455.2 and 462.1 eV. For the Al 2p region (Figure 3b), two peaks were observed. The main peak at 74.2 eV with a high area fraction corresponded to the AlN compound. The minor component at about 75.8 eV corresponds to aluminum oxide ($AlO_x$). For the Si 2p region (Figure 3c), the silicon binding energy spectrum was also divided into two peaks. The major peak Si 2p component at about 101.6 eV corresponded to silicon nitride (i.e., $SiN_x$), and a very weak peak around 103.7 eV corresponding to silicon oxide ($SiO_x$) was observed. Since no crystalline $Si_3N_4$ phase can be identified in the SAED pattern (Figure 2b), it is evident that the $Si_3N_4$ phase in the coatings is in an amorphous state. This result is also in good agreement with other early studies [5,8]. The TiN, AlN and $SiN_x$ compounds were observed for the N 1s spectrum (Figure 3d). The existence of $AlO_x$, $TiO_x$ and $SiO_x$ compounds, in the Figure 3e, was also detected. The peak of $TiO_x$ (or CuO) would be overlapped by two peaks of $TiO_x$ and $CuO_x$ because the binding energies are too close for the O 1s spectrum. Finally, for the Cu 2p region (Figure 3f), two peaks located at 952.3 and 932.8 eV were detected. The major peak Cu 2p component at 932.8 eV corresponded to pure Cu, and a relatively smaller

peak at 952.3 eV corresponding to non-stoichiometric $CuO_x$ was characterized. On the other hand, Shi et al. reported [33] that the incorporated copper atoms exist individually as small sized Cu particles instead of combining with other atoms to form any other chemical bonding (e.g., Cu-N, Cu-O, etc) within the $Ti_{0.43}Al_{0.48}Si_{0.06}Cu_{0.03}N$ coatings. Combined with the TEM results (Figure 2), it can be concluded that the incorporated Cu atoms were presented as a phase of amorphous copper oxide ($CuO_x$) or very small nanosized Cu, smaller than 1 nm, in the grain-boundary region.

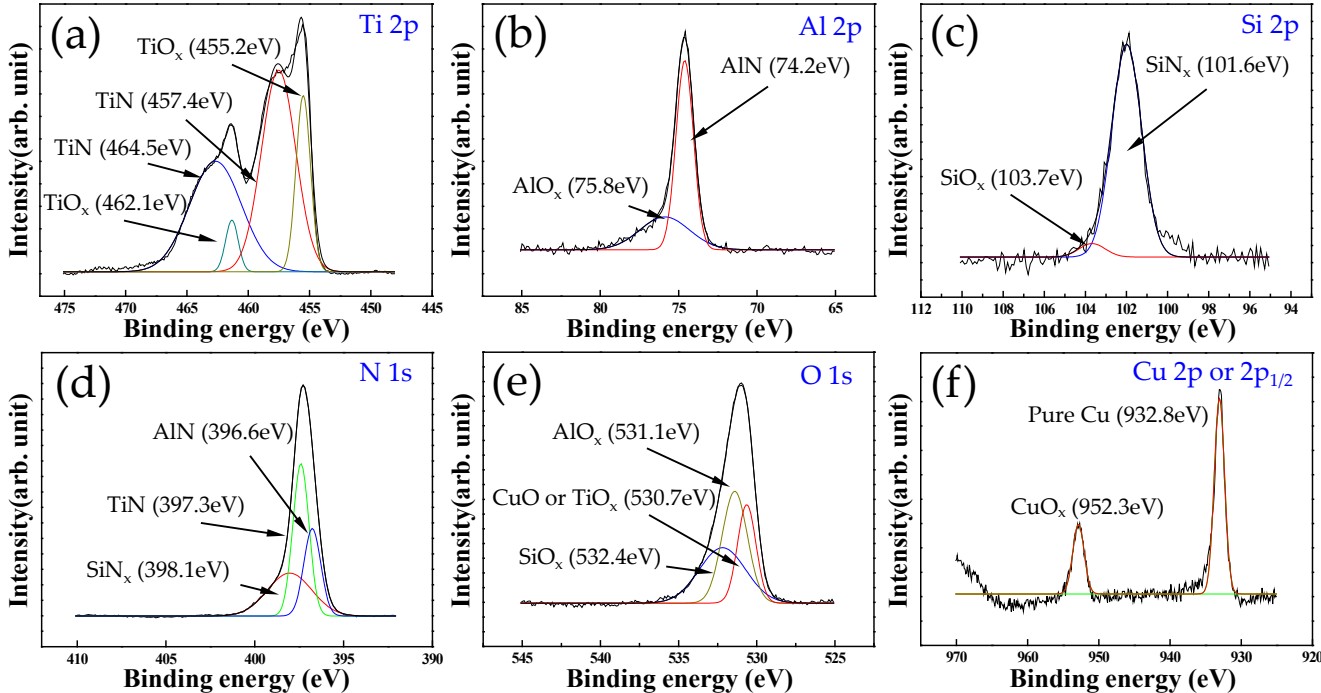

**Figure 3.** XPS spectrum of (**a**) Ti 2p, (**b**) Al 2p, (**c**) Si 2p, (**d**) N 1s, (**e**) O 1s, and (**f**) Cu 2p or $2p_{1/2}$ for the TiAlSiN–Cu(2.92 at.%) coating.

Atomic force microscopy (AFM) was carried out to observe the surface and to determine the surface roughness of the TiAlSiN–Cu coatings with various Cu contents, as shown in Figure 4. As the Cu content increased, the surface roughness of the TiAlSiN–Cu coatings gradually decreased from 80.2 to 54.4 nm. This result can be coupled with the TEM analysis shown in Figure 2, in that segregated amorphous phases in the TiAlSiN–Cu coatings resulted in a grain size refinement of (Ti,Al)N crystallites during coating growth. Therefore, the nano-sized crystallites with regular and spherical shapes affected the surface morphology of the TiAlSiN–Cu coatings. In addition, according to a previous report by Lin et al. [8], when high ion energy and ion flux bombardment are used in the coating growth, the grain size and the surface roughness of the films can be reduced. The morphology of the TiAlSiN–Cu(1.01 at.%) coating indicates relatively large and stubby grooves with separated grains. This would be related to a relatively large average grain size and porous structure with a surface roughness of about 80.2 nm. On the other hand, the coatings with contents above 2.92 at.% Cu have short and sharp grooves with smaller grains, which can be related to grain size refinement due to more amorphous volume fraction in the coatings, thereby showing a more dense microstructure (Figure 4c,d).

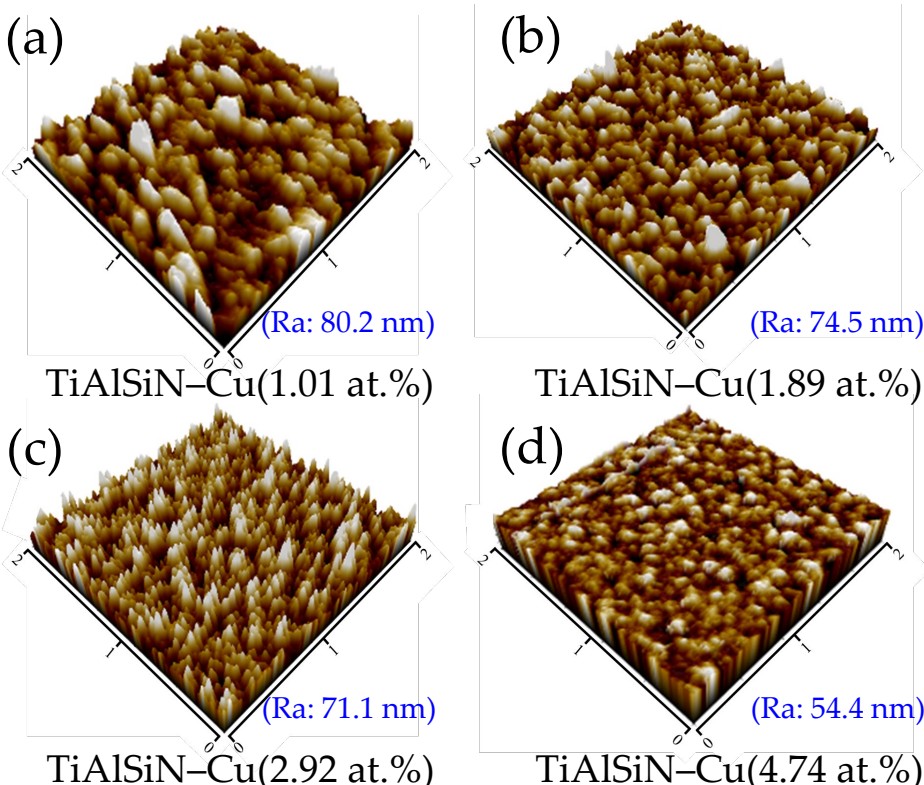

**Figure 4.** Three-dimensional AFM surface morphologies and their surface roughness of (**a**) TiAlSiN–Cu(1.01 at.%), (**b**) TiAlSiN–Cu(1.89 at.%), (**c**) TiAlSiN–Cu(2.92 at.%), and (**d**) TiAlSiN–Cu(4.74 at.%) coating deposited on a silicon wafer substrate.

Based on the results from TEM, XPS and AFM analyses, it is concluded that the TiAlSiN–Cu coatings have a nanocomposite microstructure consisting of (Ti,Al)N nano-crystallites embedded in an amorphous matrix, which is a mixture of the $TiO_x$, $AlO_x$, $SiO_x$, $SiN_x$, and $CuO_x$ phases.

### 3.2. Mechanical and Tribological Properties of the Coatings

Nanohardness, Young's modulus, H/E* ratio, and $H^3/E^{*2}$ value of the TiAlSiN–Cu coatings as a function of Cu content were plotted in Figure 5. As shown in Figure 5a, the nanohardness of the TiAlSiN–Cu coatings decreased from ~47 GPa at 1.01 at.% to ~34 GPa at 4.74 at.% Cu content. The nanohardness was almost constant at about 45 GPa ranging from 1.89 to 2.92 at.% Cu content and decreased again with a further increase in Cu content to about 34 GPa at a Cu content of about 4.74 at.%. The TiAlSiN–Cu coatings with Cu content from 1.01 at.% to 2.92 at.% exhibited a superhardness (>40 GPa) of about 45 GPa. The reason for maintaining the superhardness of the TiAlSiN–Cu coatings with a small amount of Cu is most likely the result of grain boundary hardening, created by the strong cohesive energy of inter-phase boundaries in terms of the Griffith criterion [34] and Hall-Patch relation derived from grain size refinement [35]. It would also result from the uniform distribution of the (Ti,Al)N nanocrystallites embedded in an amorphous matrix (e.g $SiN_x$, $CuO_x$, etc), as characterized in the microstructure section (Figures 2, 3, and 4). With the further increasing of Cu content, even though the grain size further decreased, the nanohardness and Young's modulus suddenly decreased. The decrease in the hardness and Young's modulus of the coatings appears to be affected by increasing amorphous volume fraction in the TiAlSiN–Cu coatings.

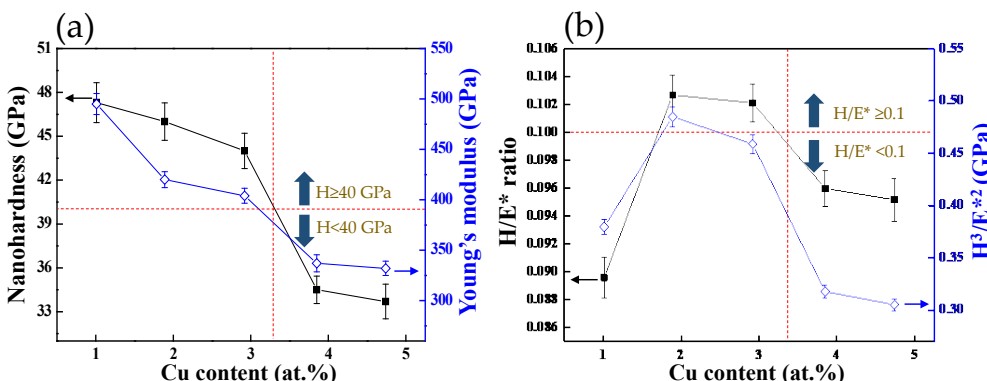

**Figure 5.** (**a**) Nanohardness (H) and effective Young's modulus (E*) and (**b**) H/E* ratio and $H^3/E^{*2}$ value of the TiAlSiN–Cu coatings as a function of Cu content.

In addition, H/E* ratio (so-called 'elastic strain to failure') and $H^3/E^{*2}$ value (the so-called 'resistance of materials against plastic deformation') were calculated from the obtained hardness (H) and effective Young's modulus (E*). They are considered as good indicators in the determination of the resistance of coatings to cracking and wear [8]. The H/E* ratio, in particular, is often proposed as a key parameter to indirectly estimate the wear resistance and fracture toughness of the hard and tribological coatings [13,36]. As the Cu content increased, the H/E* value of TiAlSiN–Cu coatings also increased from ~0.090 to 0.102. The H/E* value was also almost constant at about 0.102, ranging from 1.89 to 2.92 at.% Cu content and decreased again with a further increase in Cu content to about 0.095 at 4.74 at.% Cu content. The TiAlSiN–Cu coatings with Cu content from 1.89 to 2.92 at.% exhibited a maximum value of about 0.102. Musil et al. have reported correlations between mechanical properties (H, E*, $W_e$ and H/E ratio) on their Al–Cu–O nanocomposite coatings [13]. The Al–Cu–O coatings with H/E* ≥ 0.1, which is a critical value to show an enhanced resistance of coating to cracking, had a highly elastic recovery without any cracks after the diamond indenter load test. However, the coatings with H/E ≤ 0.1 exhibited relatively low elastic recovery with many cracks on the corner of the indenters. From the results in Figure 5, it can be suggested that the TiAlSiN–Cu coatings with a Cu content of 1.89 to 2.92 at.% will show better wear resistance and fracture toughness.

Furthermore, in order to counteract brittleness and improve fracture toughness with a high adhesion strength between the coatings and substrates, a functionally graded coating's (FGC's) architecture (Ti/TiN/TiAlN/TiAlSiN/TiAlSiN–Cu multi-layers) was applied to all TiAlSiN–Cu coatings in this work. Figure 6 shows critical loads and their optical micrographs of the scratch tracks for the TiAlSiN–Cu coatings on WC–Co substrates with various Cu contents. The adhesion strength between hard coatings and substrates plays an important role in the coating performance in industrial applications. Stallard et al. reported [37] that the adhesive failure strength can be classified into four modes ($L_{C1}$ to $L_{C4}$) in the scratch tests under a progressive load. In general, the adhesive failure mode $L_{C2}$ is often used for determining adhesion strength. In Figure 6, when the Cu content increased from 1.01 to 2.92 at.%, the adhesion strength ($L_{C2}$) of the TiAlSiN–Cu coatings increased slightly from 49 to 60 N, and then decreased again to around 40 N when Cu content increased to 5.0 at.%. The improvement of adhesion strength with the variation of Cu content would mainly be attributed to intrinsic stress release due to the soft metallic Cu and amorphous $CuO_x$ phase in the grain boundary during coating growth with adhesion and supporting layers. It was also reported that the improvement in adhesion strength of the hard coatings can be attributed to the combined effects of H/E ratio and appropriate compressive residual stress [38]. Similar results were also found in other's studies [22,30].

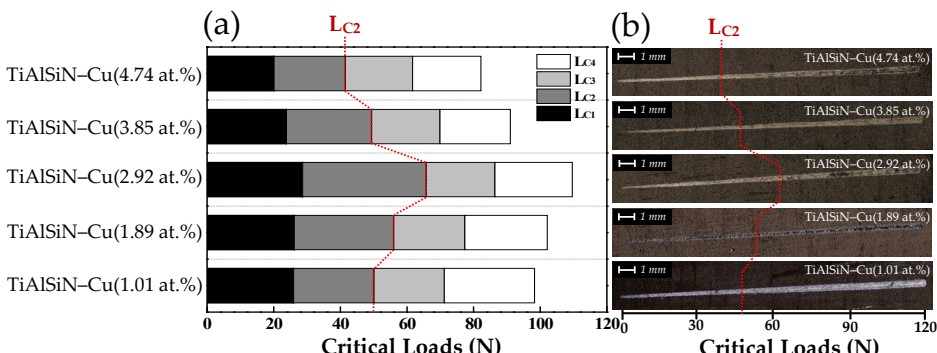

**Figure 6.** (**a**) Critical loads and (**b**) their optical micrographs of the scratch tracks for the TiAlSiN–Cu coatings on WC–Co substrates with various Cu contents.

Figure 7 provides the friction coefficient of the TiAlSiN–Cu coatings with various Cu contents against an Inconel ball. The friction coefficient of the TiAlSiN–Cu coatings gradually decreased by increasing the Cu content, and showed a minimum value of approximately 0.46 at a Cu content of 2.92 at.%, and then rebounded with the further increase in the Cu content above 4.74 at.%. The improvement of friction coefficient in the TiAlSiN–Cu coatings with 2.92 at.% Cu is most likely caused by a very smooth surface morphology, as shown in AFM images (Figure 4c), due to the grain size refinement with nanocrystals of 5–7 nm in size embedded in the amorphous matrix. Another possible reason is related to the formation of smooth solid-lubricant layers formed by tribo-chemical reactions during the sliding tests. For example, soft metallic copper compounds such as pure Cu particles and amorphous $CuO_x$ in the coatings react with ambient $H_2O$ and oxygen to produce $CuO_x$ or $Cu(OH)_x$ tribo-layers. These tribo-oxides, as a good potential solid lubricant, have been used to effectively reduce friction and wear at extreme conditions [39]. Copper oxide (CuO, $T_m$: 1326 °C) is reported to be a softened oxide [40] and to be more easily sheared than the metals and ceramic nitrides [41]. Yao et al. found [42] that the incorporation of the CuO compound, which was formed on worn surfaces by a tribo-chemical reaction at 800°C, can reduce the friction coefficient to about 0.16. Figure 7b shows the optical micrograph of wear track of the TiAlSiN–Cu coatings with various Cu contents. The surface morphologies of the wear tracks for the TiAlSiN–Cu coatings with Cu content from 1.89 to 2.92 at.% were smooth, and the width of the wear tracks were narrow. From a practical view, these morphologies probably will result in a good wear resistance of the coatings. On the other hand, the surface morphology for the TiAlSiN–Cu(4.74 at.%) coating was relatively rough and the width of the wear tracks was wide. This result indicates that the relatively soft TiAlSiN–Cu coatings with low H/E* value have an abrasive wear behavior.

Figure 8 represents the coating thickness, wear depth, and wear rate of the TiAlSiN–Cu coatings after the wear sliding tests. The different wear depths of the TiAlSiN–Cu coatings could be due to the different friction and wear behaviors, which could directly influence the wear rate of the coatings, as shown in Figure 8b. The wear rate of the TiAlSiN–Cu coatings was slightly decreased from $13 \times 10^{-6}$ to $10 \times 10^{-6}$ mm$^3$/(N·m) with the increasing of the Cu content from 1.01 to 2.92 at.%, which would be due to the initial increase of Cu content in the coatings and the improvement of mechanical properties (H, H/E*, and $H^3/E^{*2}$). Pappacena et al. reported [21] that the wear rate also can be correlated to the adhesion energy and the residual stress with the initial copper content in their MoN/Cu composite coatings. In addition, the improvement in wear resistance of the TiAlSiN–Cu(1.01 to 2.92 at.%) coatings could be attributed to the increase in the lubricious oxides of $CuO_x$, $AlO_x$, and $SiO_x$ formed during tribo-oxidation, which usually leads to lower friction coefficients and wear resistance [40]. On the other hand, at the Cu content of above 3.85 at.%, the wear rate of the TiAlSiN–Cu coatings steeply increased to about $25 \times 10^{-6}$ mm$^3$/(N·m). This large increase in wear rate is due to the abrasive wear behavior between the relatively

soft coating with lower hardness (~34 GPa) and the Inconel ball, and is also due to the diminishment of the mechanical properties of the hardness, H/E* ratio, and $H^3/E^{*2}$ [8].

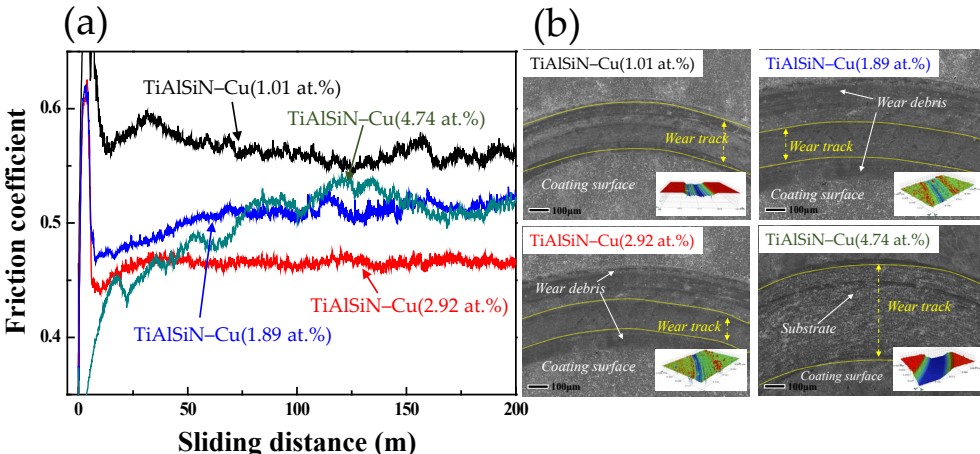

**Figure 7.** (**a**) Friction coefficient and (**b**) their SEM and 3D profiling image of the wear track for the TiAlSiN–Cu coatings with various Cu contents against the Inconel ball. (Test conditions: Load, 3 N; ball dia., 3 mm; track dia., 7 mm; linear sliding speed, 100 rpm; room temp.).

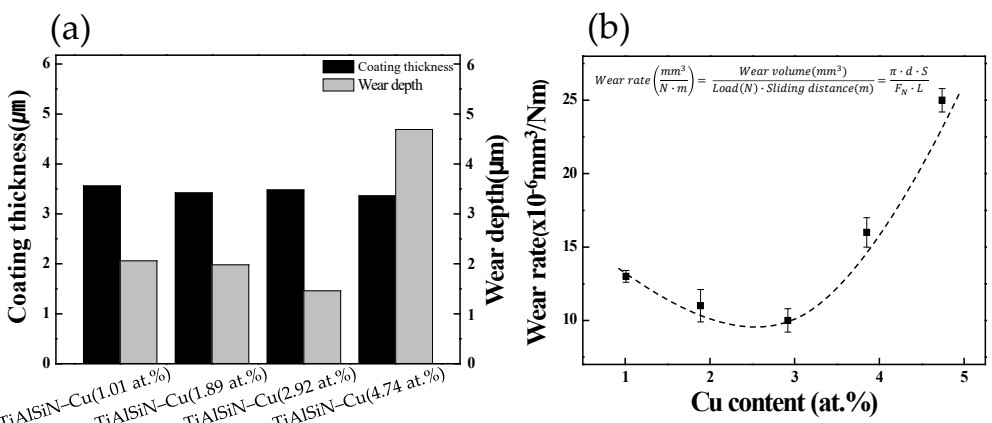

**Figure 8.** (**a**) Coating thickness, wear depth, and (**b**) wear rate of the TiAlSiN–Cu coatings as a function of Cu content after sliding tests.

Therefore, based on the results of nanohardness, Young's modulus, H/E*, $H^3/E^{*2}$, and adhesion strength of the coatings, the TiAlSiN–Cu coatings with Cu content up to about 3 at.% could provide superior wear resistance with higher fracture toughness than those of the TiAlSiN–Cu coatings with Cu contents above 4 at.%.

## 4. Conclusions

In this work, functionally graded TiAlSiN–Cu coatings with different Cu contents were fabricated by a filtered cathodic arc ion plating system using Ti, $TiAl_2$, $Ti_4Si$, and $Ti_4Cu$ alloying targets in an argon-nitrogen atmosphere. The microstructural, mechanical and tribological properties were investigated with various characterizations. The following conclusions were observed:

(1) It was revealed that the TiAlSiN–Cu coatings were a nanocomposite consisting of nano-sized (Ti,Al)N crystallites (~5 to 7 nm in size) embedded in an amorphous matrix, which is a mixture of the $TiO_x$, $AlO_x$, $SiO_x$, $SiN_x$, and $CuO_x$ phase. The addition of Cu atoms into the TiAlSiN coatings led to the formation of an amorphous copper oxide ($CuO_x$) phase in the amorphous matrix.

(2) The TiAlSiN–Cu coatings with Cu content up to about 3 at.% presented high hardness (~46 GPa), high $H/E^*$ (~0.102) and $H^3/E^{*2}$ (~0.50 GPa) values regarding the coating's fracture toughness, and excellent adhesion strength ($L_{C2}$, ~60 N).

(3) The addition of Cu atoms also improved the tribological property and wear resistance. The friction coefficient of the TiAlSiN–Cu coatings gradually decreased by increasing the Cu content and showed a minimum value of ~0.46 at a Cu content of 2.92 at.%. The formation of a copper oxide ($CuO_x$) phase during coating growth or a sliding test played a key role as smooth solid-lubricant layers, and reduced the average friction coefficient (~0.46) and wear rate ($\sim 10 \times 10^{-6}$ $mm^3/N·m$).

(4) Such a good combination of mechanical and tribological properties of the TiAlSiN–Cu coatings with Cu content up to about 3 at.% would indicate the considerable potential of the coatings for applications in mechanical components. However, further studies are necessary to investigate the oxidation properties of these coatings.

**Author Contributions:** Conceptualization, S.-B.H. and I.-W.P.; methodology, S.-B.H. and J.-H.K.; validation, W.R.K. and S.-H.C.; formal analysis, W.R.K. and S.-H.C.; resources, J.-H.L.; writing—original draft preparation, S.-B.H.; writing—review and editing, I.-W.P. and D.K.; project administration, J.-H.K. and J.-H.L.; supervision, I.-W.P. and D.K. All authors have read and agreed to the published version of the manuscript.

**Funding:** This work was supported by the Renewable Energy Program of the Korea Institute of Energy Technology Evaluation and Planning (KETEP) grant funded by the Korea government Ministry of Trade, Industry and Energy (MOTIE) (Grant No. 20203030040060, Development of ultra-high pressure hydrogen piping valve (105 MPa), dispenser, heat exchanger, and monitoring system with hydrogen embrittlement suppression effect and leak detection function).

**Institutional Review Board Statement:** Not applicable.

**Informed Consent Statement:** Not applicable.

**Data Availability Statement:** Not applicable.

**Conflicts of Interest:** The authors declare that they have no conflicts of interest.

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
