# Peer review of "Effects of Copper Content on the Microstructural, Mechanical and Tribological Properties of TiAlSiN–Cu Superhard Nanocomposite Coatings"

_coatings, doi:10.3390/coatings12121995_

Round 1

Reviewer 1 Report

 This manuscript described fabricating graded TiAlSiN–Cu coatings with various copper (Cu) contents by a filtered cathodic arc ion plating technique using four different (Ti, TiAl2, Ti4Si, and Ti4Cu) targets in an argon-nitrogen atmosphere. The manuscript is well organized and presented. I just have two suggestions:

1. Please supply the XRD results, so that the readers can have the idea of coating phase structures. 

2. Please supply the detailed wear track profiles and SEM images in the wear tracks. As we know, the Inconel alloy is much softer than the nitride coatings. What is the mechanism of coating wear and friction during the wear tests? What are the differences in the coatings with different Cu contents? More experimental proofs and discussion is needed. 

Reviewer 2 Report

To coat cutting tools with adequate coatings is a well-known approach to improve their wear resistance. The present article contains valuable experimental data about microstructure, mechanical and tribological properties of functionally graded TiAlSiN–Cu coatings with various copper contents which were deposited on two substrates: silicon wafers and WC–Co substrates. However, some comments could be made.

 2) Experimental procedures

2.1. Coating deposition

a) at p. 2, line 87, the authors wrote that ” TiAlSiN–Cu coatings were deposited onto silicon wafers and WC–Co substrates”; the authors should show why used two substrates, and how each experimental coating model was used to characterize the superhard nanocomposite coatings.

 3) Results and discussion

b) in Figure 1a, the authors showed that a functionally graded TiAlSiN–Cu coatings with multilayers architecture were deposited on two substrates: Ti/TiN/TiAlN/TiAlSiN/TiAlSiN-Cu; in Table 1, Figure 3 and discussion about these results, the authors stated ” that the TiAlSiN–Cu coatings have a nanocomposite microstructure consisting of (Ti,Al)N nano-crystallites embedded in an amorphous matrix, which is a mixture of TiOx, AlOx, SiOx, 250 SiNx, and CuOx phase.” (p. 5-7); having in view that the authors showed that ”Oxygen source was delivered from the target or chamber as an impurity”, the following question which can be put: the oxides were formed in each layer of the graded coatings or only in the last layer (TiAlSiN-Cu)? Also, the experimental conditions of the coating process allow as the all oxides to be formed in this high concentration argon-nitrogen atmosphere?

c) the authors should show how much areas were investigated and if the investigations were made only on the surface of the coatings or they made both on the surface and transversal section; the authors identified compositional, microstructural, mechanical and tribological differences between the investigated zones?

Reviewer 3 Report

Attached

Reviewer 4 Report

1. Include more discussion about nano-composites coating in the introduction section. Paper related to the addition of copper for coatings. 

2. Explain in detail. Why initially with an increase in copper content, Young's modulus increased but reduced at a higher percentage of copper. 

3. Friction coefficient in the range of 0.4-0.6 for coating is too much. What are the intended applications for this coating?

4. Explain in detail, why there is a sudden increase in wear rate with an increase in copper content after 3 %?

5. Authors are recommended to simulate the coating with any FEA package and test it for COF using similar parameters to verify the findings. 

Round 2

Reviewer 2 Report

The revised form includes a few new details on the literature, methods and the discussion which provide a clearer understanding on the relationship between processing, coatings features and their performance.

Reviewer 4 Report

The authors have modified the paper. It can be accepted in its present form.